# Termbot: A Chatbot-Based Crossword Game for Gamified Medical Terminology Learning

**DOI:** 10.3390/ijerph20054185

**Published:** 2023-02-26

**Authors:** Mei-Hua Hsu, Tien-Ming Chan, Chi-Shun Yu

**Affiliations:** 1Center for General Education, Chang Gung University of Science and Technology, Taoyuan City 33303, Taiwan; 2Chang Gung Memorial Hospital, Chang Gung University, Taoyuan City 33303, Taiwan; 3Department of Electrical Engineering, National Taipei University of Technology, Taipei City 10608, Taiwan

**Keywords:** medical terminology, chatbot, M-learning, crossword puzzles, line

## Abstract

Medical terminology can be challenging for healthcare students due to its unfamiliar and lengthy terms. Traditional methods such as flashcards and memorization can be ineffective and require significant effort. To address this, an online chatbot-based learning model called Termbot was designed to provide an engaging and convenient method for enhancing medical terminology learning. Termbot, accessible through the LINE platform, offers crossword puzzles that turn boring medical terms into a fun learning experience. An experimental study was conducted, which showed that students who trained with Termbot made significant progress in learning medical terms, demonstrating the potential of chatbots to improve learning outcomes. Termbot’s gamified approach to learning can also be applied to other fields, making it a useful tool for students to learn medical terminology conveniently and enjoyably.

## 1. Introduction

Medical terminology is a crucial aspect of clinical thinking and application, and its learning process can be challenging and frustrating for many medical professionals and students. However, familiarizing oneself with medical terminology can be immensely beneficial, not just for medical students but also for the general public. It can help one prepare better before visiting the hospital and make it easier to communicate with medical professionals. For example, it can aid in determining which outpatient department to visit and what type of specialist doctor to consult. Moreover, knowing medical terms can also help individuals engaged in related careers, such as writing medical articles, translating medical content, or working in medical insurance. It can give a clear understanding of one’s health problems and help medical professionals to diagnose and treat them effectively. In conclusion, while the learning process of medical terminology can be challenging, the benefits it offers make it a valuable investment of time and effort. By familiarizing oneself with the basic medical terms, one can take an active role in their healthcare and improve their overall health and well-being.

With the widespread availability of smartphones, mobile learning has become a popular trend, allowing learners to study at their convenience anytime and anywhere. The oral aspect of mobile learning has been noted as particularly important [1]. Digital games and other technologies have also been used to enhance learning, such as Troussea et al.’s mobile language learning application with a chatterbot, MACE: a mobile artificial conversational entity for adapting domain knowledge and generating personalized advice [2,3,4], and Marougkas’ framework for personalized virtual reality learning environments [5]. Studies have shown that students prefer instant messaging as a communication tool [6], and a survey of college students revealed they spend nearly 6 h a day on their smartphones [7]. To meet the learning habits of today’s students, an online chatbot was designed with crossword puzzles to help students learn medical terminology. Digital literacy has become increasingly crucial in education and should be prioritized, as technology continues to advance rapidly. The online chatbot learning method provides an interactive and engaging way to learn medical terminology, as turning these terms into games can improve motivation and make learning more interesting. This method is also not limited by time and space, allowing users to learn anywhere and anytime.

The crossword puzzle strategy is a form of active learning that engages students by having them fill in crossword games created by the teacher. According to Kazimoglu, digital games foster motivation by providing players with solutions that allow them to progress to the next level, contributing to the growth of the video game industry [8]. These games can come in various forms, including educational games, adventure games, action games, and role-playing games. Atkinson [9] suggested that incorporating puzzles into teaching can enliven classroom activities, providing a fun and challenging aspect to learning.

Kumar et al. emphasized that, while online crosswords may not be a substitute for traditional crosswords, they still offer valuable teaching opportunities, as they can pique students’ interest and encourage them to learn [10]. Bailey et al. suggested that crossword puzzles can create a friendly and competitive environment, promoting student interaction and engagement [11]. Gaikwad & Tankhiwale believed that crossword puzzles can stimulate thinking, increase vocabulary, and help foster critical thinking skills [12].

Patel and Dave observed that students appreciated the use of crossword puzzles in their learning, as it allowed them to engage in important learning activities. The strategic use of crossword puzzles can simplify the delivery of content and provide opportunities for students to discuss and review fundamental concepts in their university studies [13].

Crossword puzzles are an effective tool for memorizing vocabulary and promoting active learning and critical thinking skills.

Research question: Can the use of Termbot aid in improving students’ medical terminology?

## 2. Related Works

Chatbots are computer programs that simulate human conversation using text or speech recognition. They are designed to make users believe they are communicating with a real person in real time. Advancements in technology have enabled chatbots to employ advanced artificial intelligence algorithms and natural language processing to provide intelligent and meaningful responses. The early chatbots relied on simple pattern matching and string processing, but modern systems have evolved to use more sophisticated knowledge-based models to generate human-like responses. Especially during COVID-19, chatbots have played an important role, as in [14]. It is a tool that uses machine learning to identify and notify users of potentially inaccurate or misleading information related to COVID-19 in web search engine results. The purpose of this tool is to combat the spread of misinformation during the COVID-19 pandemic by helping users identify and avoid unreliable sources of information. Recently, a survey conducted by [15] reviewed the existing literature on healthcare chatbots related to COVID-19. The survey identified and characterized these emerging technologies and their applications for combating COVID-19 and described the challenges associated with their implementation. Chatbots are becoming increasingly popular in various fields such as medicine, commerce, and education, thanks to their ability to interact with users via text or text-to-speech conversion. With advancements in AI and natural language processing, chatbots can now generate more human-like responses, tricking users into thinking they are talking to a real person. Winkler and Söllner noted the growing use of chatbots in different industries [16]. The use of chatbots in education has been shown to have a positive impact on learning outcomes. In large classes with over 100 students per instructor, chatbots can provide individualized support, which can be beneficial for students. Hussain conducted a survey that looked into the different types of chatbots and the conversational context-handling methods of early and modern chatbots [17]. In distance learning, chatbots can be a valuable tool in understanding student interaction and task completion. Heller found that Freudbot, a chatbot designed for psychology students, resulted in high interaction records and task execution rates in a distance learning environment [18]. Matsuura and Ishimura also conducted a study comparing the use of chatbots and humanoid robots in science lectures and found that the use of chatbots improved students’ understanding of the material [19]. According to Nurhayati et al., chatbots play a crucial role in facilitating the learning process and improving learning outcomes [20].

To make learning medical terminology more engaging and effective, this study presents a novel approach that uses JavaScript, jQuery, and HTML to develop a chatbot system with crossword puzzles, named Termbot. Termbot can be accessed via the LINE platform and is aimed at providing students with a convenient and interactive way to practice and memorize English medical terminology.

LINE, a freeware app for instant communication on various electronic devices, is widely used in Taiwan, including by students. Termbot leverages this popular platform to offer an entertaining and effective learning experience for students. As students complete the crossword puzzles, Termbot records their incorrect answers and categorizes the students into different groups, allowing the teacher to provide targeted remedial instruction based on the needs of each student. This unique feature sets Termbot apart and represents a significant contribution of this study to the field of medical terminology education.

## 3. Method

### 3.1. Study Design

This study employs an experimental method to assess the impact of Termbot on students’ learning of medical terminology. The sample participants were divided into two groups: the experimental group and the control group. The experimental group received instruction on the use of Termbot, while the control group received traditional teaching methods. The researchers compared the results between the two groups to determine the effectiveness of Termbot on students’ academic performance.

### 3.2. Participants

Sixty participants were recruited for this two-month study, with 30 anonymous nursing students assigned to the experimental group and 30 anonymous nursing students assigned to the control group. The experimental group was given access to Termbot for practicing medical terminology at any time, while the control group did not have access to Termbot. After two months, all 60 participants took a posttest for the online medical terminology exam.

### 3.3. Study Instruments

Termbot, a chatbot based on crossword puzzles and developed using Python, has been introduced in this study. It is designed to be used on the LINE platform, which is known for its user-friendliness and diverse messaging options, including flex messages. LINE is a popular instant messaging app that provides a wide range of functions, such as shopping, mobile payments, news, and live broadcasts.

The Termbot system comprises two major modules, namely the database and the processing engine, as illustrated in Figure 1. Users can access Termbot through LINE on their mobile devices, which run on either the iOS or Android operating system.

To set up Termbot on the server side, Python version 3.8.13, Ngrok, and a LINE Developers account are required. The system also uses other packages, including line-bot-sdk version 2.3.0 and flask version 2.2.2.

Termbot can be accessed on various devices, including smartphones, iPads, laptops, and personal computers, offering students a convenient and accessible way to practice their medical terminology. After the teacher inputs medical terminology and clues, the chatbot automatically generates crossword puzzles for the students to complete. The architecture of Termbot is shown in Figure 1, which is a simplified diagram of the system’s design.

### 3.4. The Database

The Termbot system stores all medical terminology in a relational database named PostgreSQL [21]. As a gamified learning tool, Termbot utilizes the open-source software PostgreSQL 15 to manage and organize its database. PostgreSQL is an open-source relational database management system similar to other proprietary systems such as MS SQL and Oracle. However, unlike these systems, PostgreSQL is available as a free and open-source software. Figure 2 illustrates the relationships within the database.

The fields of the tables are described as follows:

The “Student” data sheet contains all the relevant information about the students, including their unique identifier “student_id” and their name recorded in the “name” column.

The “Teacher” data sheet records information about teachers, including their unique identifier “teacher_id” and the class they teach, recorded in the “class” column.

The “Vocabulary” data sheet records information about the vocabulary used in the class. The “book” column indicates which book the vocabulary is from, and the “lesson” column indicates the specific lesson in which the vocabulary is used.

The “Puzzle” data sheet records information about crossword puzzles created for the class. The “timestamp” column records when the puzzle was created, while the “auto_manual” column indicates whether the vocabulary used in the puzzle was sourced from the books or typed in manually by the teacher. A value of 0 in the “auto_manual” column indicates that the vocabulary was sourced from the books, while a value of 1 indicates that the vocabulary was typed in manually by the teacher.

The “Homework” data sheet connects puzzles with students. The “puzzle_id” column is used to identify the puzzle, while the “class_id” column identifies which classes the puzzle is related to. The “valid” column indicates whether the puzzle is still valid or not.

The “Wrong” data sheet records information about student mistakes. The “wrong_word” column is used to record each student’s wrong answer.

## 4. Termbot

Termbot is a chatbot designed to be used on the LINE platform, which serves as its user interface. Below are the functions offered by Termbot.

### 4.1. Login Function

Figure 3 shows the initial screen of Termbot, where the user is prompted to select whether they are a teacher or a student. This selection will determine the different system functions that will be available to the user.

### 4.2. Student Functions

Students have access to three main functions on the initial screen: “Today’s homework”, “Practice by book or lesson”, and “Help”. The “Today’s homework” function allows students to view the crossword puzzle homework assigned by their teacher, as shown in Figure 4. The “Practice by book or lesson” function enables students to select the specific textbook or lesson they want to practice independently, as illustrated in Figure 5 and Figure 6. Lastly, the “Help” function provides students with operational assistance when not working on crossword puzzles and additional hints when solving puzzles.

### 4.3. Teacher Functions

On the initial screen, the teacher has access to six main functions: “Auto-generate”, “Manual generate”, “Assign & Reassign”, “Generate report”, “Delete Puzzle”, and “Help”. The “Auto-generate” function enables teachers to select the textbook and lesson to be practiced by students, and the system will automatically generate the crossword puzzle questions and answers. The teacher then confirms the puzzle before sending it back to the students, as shown in Figure 7 and Figure 8. The “Manual generate” function allows teachers to add additional words or customize the puzzle by inputting words and related clues. The system will then automatically generate the crossword puzzle and send it back to the teacher for confirmation. The “Assign & Reassign” function allows teachers to assign crossword puzzles to other classes after creating them, as shown in Figure 9 and Figure 10. The “Generate report” function will produce a report of how many students have completed each crossword puzzle, their error rate, and other statistics. The “Delete Puzzle” function allows teachers to remove unwanted crossword puzzles. Lastly, the “Help” function provides instructions for using Termbot.

### 4.4. Crossword Puzzles Generator

The field of crossword puzzle generation has seen a growing interest among researchers in recent years. Several studies have proposed crossword puzzle generators for different applications. For example, Bonomo et al. used genetic algorithms and the wisdom of artificial crowds to generate crossword puzzles [22]. Yampolskiy & El-Barkouky [23] used the wisdom of artificial crowds algorithm to solve NP-hard problems. Ginsberg et al. presented search lessons learned from crossword puzzles [24]. Beacham et al. [25] compared different algorithms, models, and heuristics for solving crossword puzzles using two dictionaries. Port et al. [26] used a genetic algorithm and the wisdom of artificial crowds to solve solitaire battleship puzzles. Widodo [27] proposed a crossword puzzle generator using a genetic algorithm with multithreaded fitness calculations. Esteche et al. [28] proposed an automatic definition extraction and crossword generation system from Spanish news text. Pintér et al. [29] proposed automated word puzzle generation based on topic dictionaries. De Kegel & Haahr [30] surveyed the existing works in Procedural Content Generation (PCG) for puzzles and reviewed 32 methods across 11 categories of puzzles. Generating crossword puzzles is a known NP-hard problem. Based on these successful methods and ideas, this research developed an online chatbot called Termbot to help students learn medical terminology more effectively. With Termbot, teachers can easily enter preprepared terms and related clues, and the system will automatically generate crossword puzzles for students to practice.

### 4.5. Usability Testing

This study utilized usability testing to assess the effectiveness of Termbot in meeting the needs of users. Fifteen volunteers were recruited to participate in the evaluation process, which is crucial for enhancing the users’ experience [31]. The usability testing involved allowing users to interact with the product design prototypes or finished products and observe, record, and analyze their behavior and feedback to make improvements to the product.

The System Usability Scale (SUS) questionnaire was used to assess the usability of Termbot. The SUS questionnaire was created by John Brooke in 1986 and is a widely used method for quickly evaluating the usability of product system interfaces, desktop programs, and website interfaces [32]. The questionnaire was modified to consist of ten questions, with five being positive questions (1, 3, 5, 7, and 9) and five being negative questions (2, 4, 6, 8, and 10), as shown in Table 1. The SUS questionnaire scores range from 0 to 100, and a score below 68 indicates poor usability, according to Sauro [33].

Data was collected using Google Forms, and a total of 15 questionnaires were received with an average total score of 83.25, indicating that users were satisfied with the system.

## 5. Results

As previously stated, 30 students were part of the experimental group, and 30 students were part of the control group. The students in the experimental group utilized Termbot for a minimum of 2 h a week for medical terminology learning after class, while the students in the control group used traditional independent learning methods without the aid of Termbot. The results of the experiment indicate significant progress made by the students in the experimental group compared to those in the control group. The statistical analysis results are presented as follows.

### 5.1. Statistical Analysis of Pre-Experiment

In this study, an independent-sample *t*-test was conducted to compare the mean difference between the control group and the experimental group. The control group consisted of 30 individuals with an average test score of 54.23, while the experimental group consisted of 30 individuals with an average test score of 53.33, as shown in Table 2. The *t*-value obtained from the test statistic was 1.2749, and the *p*-value was 0.2074, which did not reach the significance level of α = 0.05. This means that the null hypothesis could not be rejected. The results of the analysis suggest that there was no significant difference between the performance of the control group and the experimental group.

### 5.2. Statistical Analysis of Post-Experiment

After two months of learning medical terminology, the 60 nursing students took the posttest again. The results showed that the students in the experimental group made significant progress, with an average test score of 82.69, as shown in Table 3. On the other hand, students in the control group, who learned using traditional methods, had an average test score of 61.31. The independent-sample *t*-test was used to compare the difference in the means between the two groups, and the results showed that the *t*-value was −10.5072 and the *p*-value was 0.0000, which is significant at the α = 0.05 level. Thus, the null hypothesis was rejected, and the alternate hypothesis was accepted. These results demonstrate that learning medical terminology through Termbot was effective in helping students improve their test scores compared to traditional learning methods.

The findings suggest that using Termbot to practice medical terminology is effective in improving students’ medical terminology knowledge. These results are consistent with the study’s expectations and provide evidence that the use of Termbot as an educational tool can significantly enhance students’ learning outcomes. The results highlight the potential of using technology in education and support the importance of incorporating digital resources into the classroom.

### 5.3. Analysis

Returning to our research question—Can the use of Termbot aid in improving students’ medical terminology? The answer is affirmative. Our results indicate that Termbot can indeed assist students in studying medical terminology. Figure 11 and Figure 12 depict the pre-experiment and post-experiment data of the control group and experimental group. From Figure 11 and Figure 12, it can be observed that the experimental group has made significant progress.

## 6. Discussion

The results of the satisfaction survey of the experimental group students were overwhelmingly positive, as shown in Table 3. Termbot was found to be an effective self-learning tool, helping students eliminate their anxiety about remembering medical terminology. The convenience of using Termbot on a mobile phone was highlighted by the students, who noted that they could practice medical terminology at any time. This makes Termbot an excellent assisted self-learning tool in line with the importance of effective out-of-class learning and practicing, as emphasized by [34]

The results of the analysis show that there is a significant difference between the means of the control group and the experimental group, with the experimental group having a significantly higher mean score. This is consistent with the results of the study by Seidlein et al. on Gamified E-learning in medical terminology: the TERMInator tool [35].

The control group students were motivated to use Termbot for medical terminology practice after seeing the significant progress made by the students in the experimental group. To further improve Termbot, the developers plan to refine the RBT-based evaluation system [36,37].

Chatbots have played a crucial role during the COVID-19 pandemic, particularly in combating misinformation, as demonstrated by the tool discussed in [14]. This tool utilizes machine learning to detect and alert users of potentially inaccurate or misleading information related to COVID-19 in web search engine results. Its primary objective is to assist users in identifying and avoiding unreliable sources of information, thereby curbing the spread of misinformation during the pandemic. Furthermore, a recent survey conducted by [15] reviewed the existing literature on healthcare chatbots concerning COVID-19. The survey identified and characterized these emerging technologies and their applications for combating COVID-19 and highlighted the challenges associated with their implementation. Our Termbot can also enable students to continue learning during COVID-19.

To promote the usage of Termbot, the authors recommend two strategies:

Offering incentives and rewards: Providing incentives and rewards can motivate students to use Termbot and strive for success. These incentives and rewards could include certificates, prizes, or recognition.

Providing regular feedback: Regular feedback on students’ progress can help to build their confidence and motivation to continue using Termbot. This feedback can also help to identify areas where they need to focus their attention and improve their learning outcomes.

## 7. Conclusions

In this study, Termbot, a crossword puzzle chatbot designed for online learning, was developed to provide an interactive and engaging way for students to learn English medical terminology. By incorporating the fun of crossword puzzles, students find learning medical terminology to be more enjoyable and effective. During their interactions with Termbot, students can use the provided online crossword puzzles to improve their understanding of medical terminology. The results of this study showed that students who used Termbot demonstrated significant progress in learning medical terms. This information can also be useful for teachers, who can use it to identify areas where students need additional support.

Termbot has proven to be an effective tool for enhancing student learning not just in medical terminology but in other subjects as well. This versatile application is designed for professionals, students, or anyone who wants to increase their knowledge or refresh their memory. Termbot is designed for convenient use, allowing users to access it at any time and place, whether they are on the go, at home, or at work. This study highlights the feasibility, convenience, and effectiveness of mobile learning. However, the limitation of the study is that it is difficult to control the frequency and duration of students’ use of Termbot.

In conclusion, Termbot is an excellent choice for self-learning, and future research will focus on expanding the available user interfaces and integrating them with a recommendation system to make it even more effective [38,39]. The combination of crossword puzzles and chatbot technology creates a unique and enjoyable way for students to learn and retain information.

## Figures and Tables

**Figure 1 ijerph-20-04185-f001:**
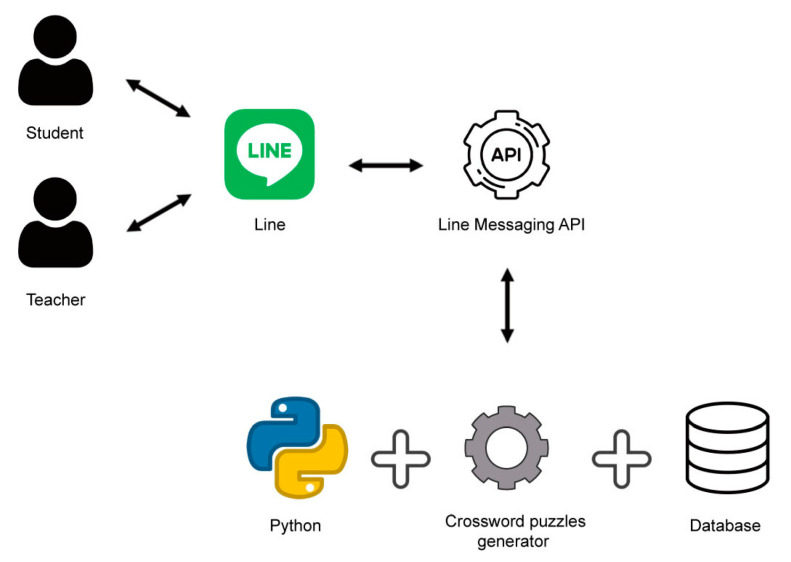
The simplified architecture of Termbot.

**Figure 2 ijerph-20-04185-f002:**
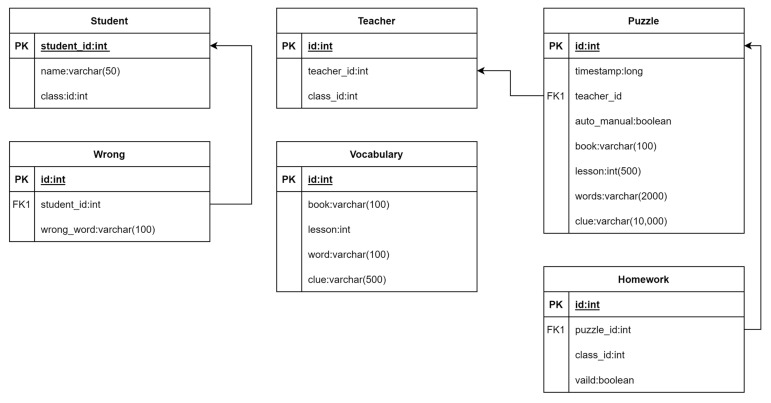
The diagram of the relationships.

**Figure 3 ijerph-20-04185-f003:**
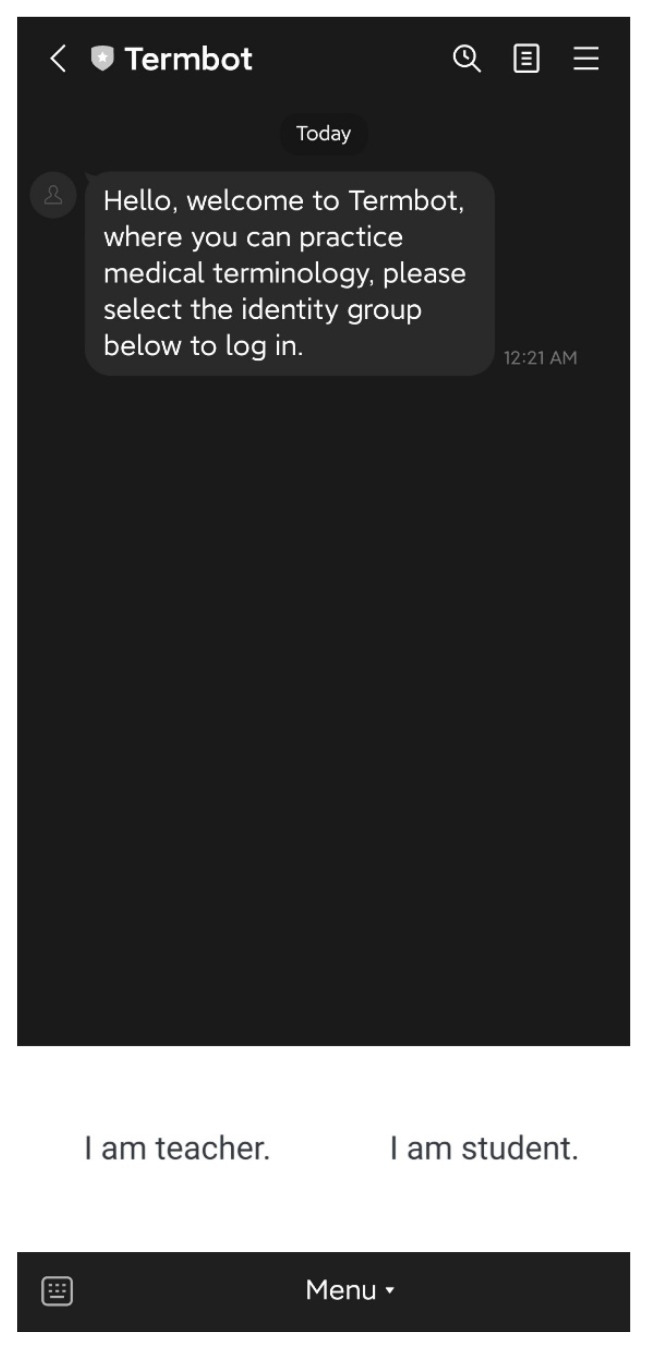
The initial screen of Termbot.

**Figure 4 ijerph-20-04185-f004:**
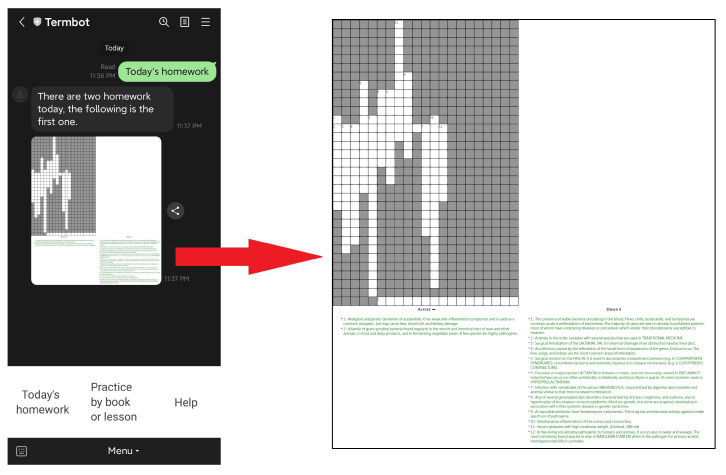
The function “Today’s homework”.

**Figure 5 ijerph-20-04185-f005:**
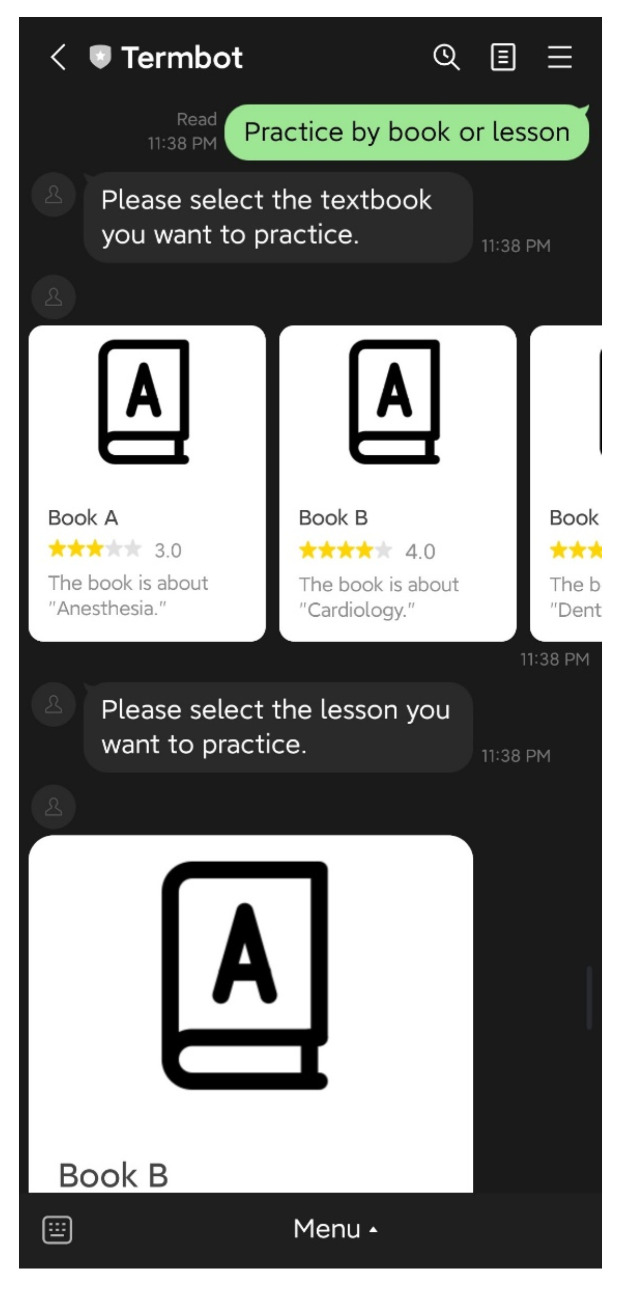
The function is “Practice by book or lesson”.

**Figure 6 ijerph-20-04185-f006:**
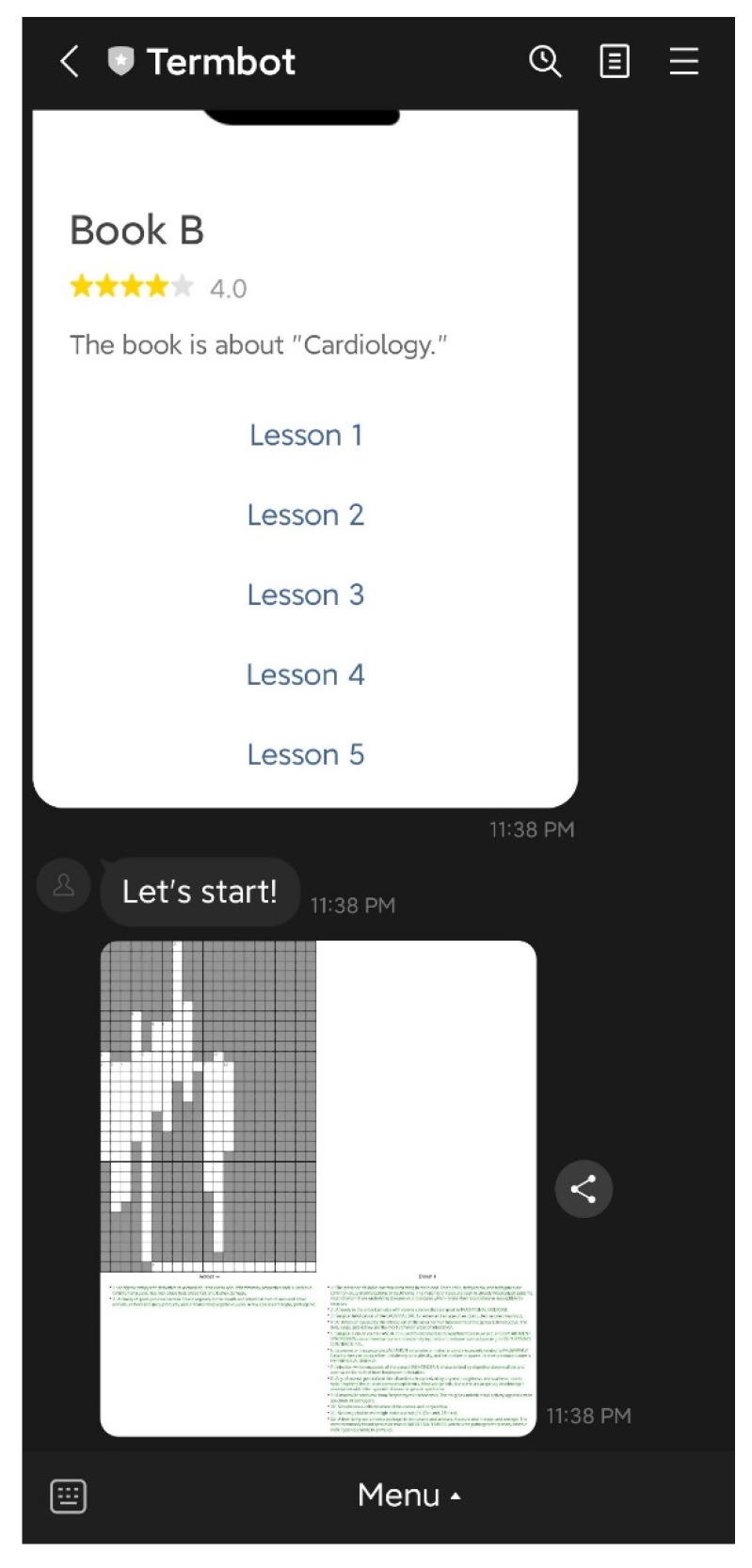
The puzzle is generated by the function “Practice by book or lesson”.

**Figure 7 ijerph-20-04185-f007:**
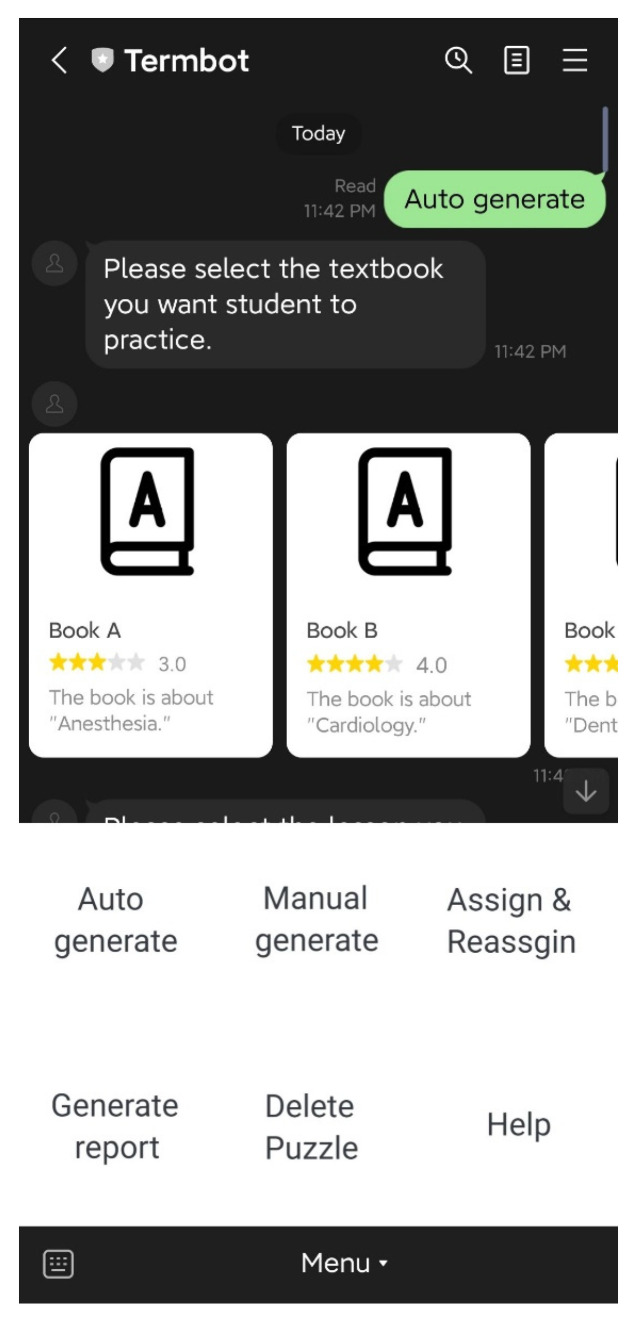
The function “Auto generate”.

**Figure 8 ijerph-20-04185-f008:**
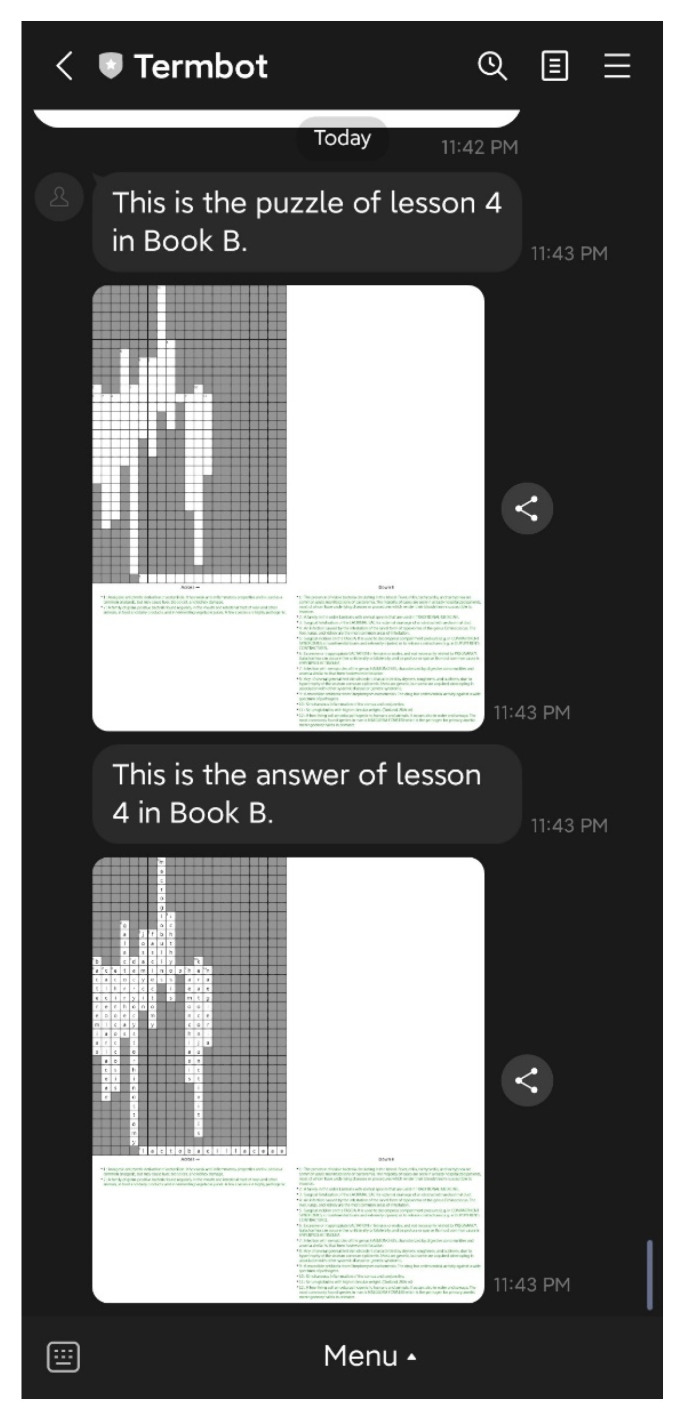
The puzzle and the answer are generated by the function “Auto generate”.

**Figure 9 ijerph-20-04185-f009:**
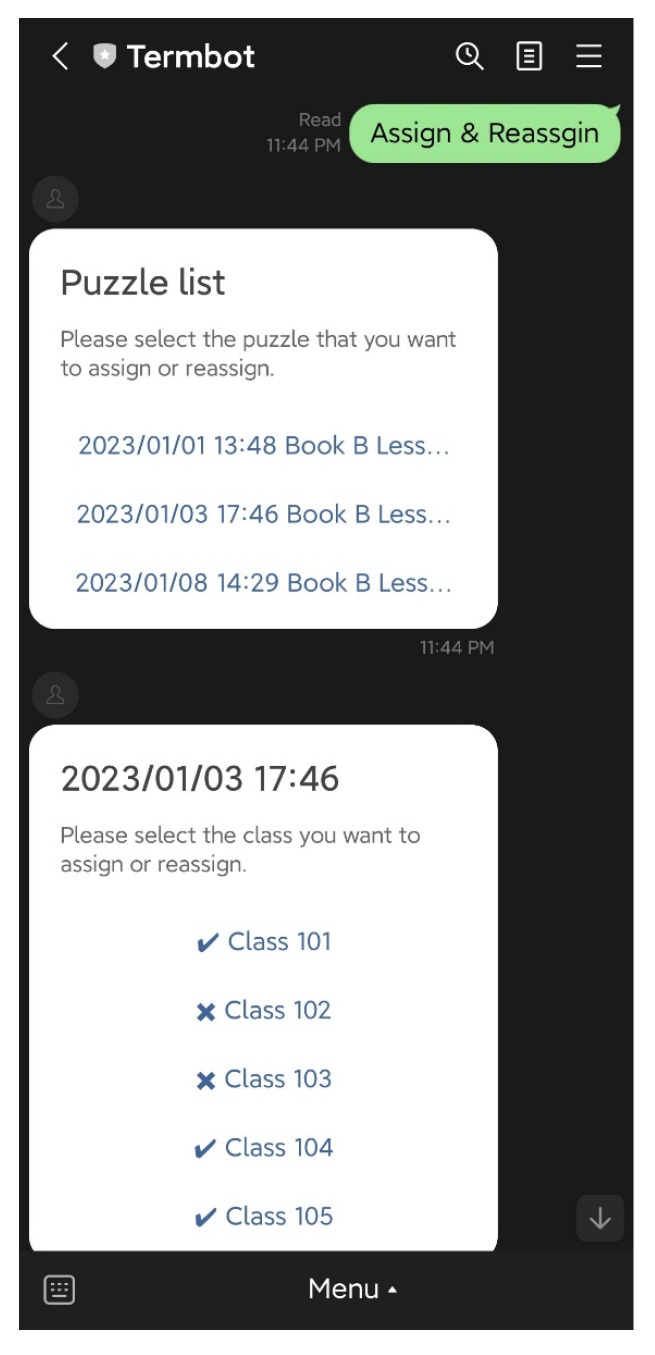
The function “Assign & Reassign”.

**Figure 10 ijerph-20-04185-f010:**
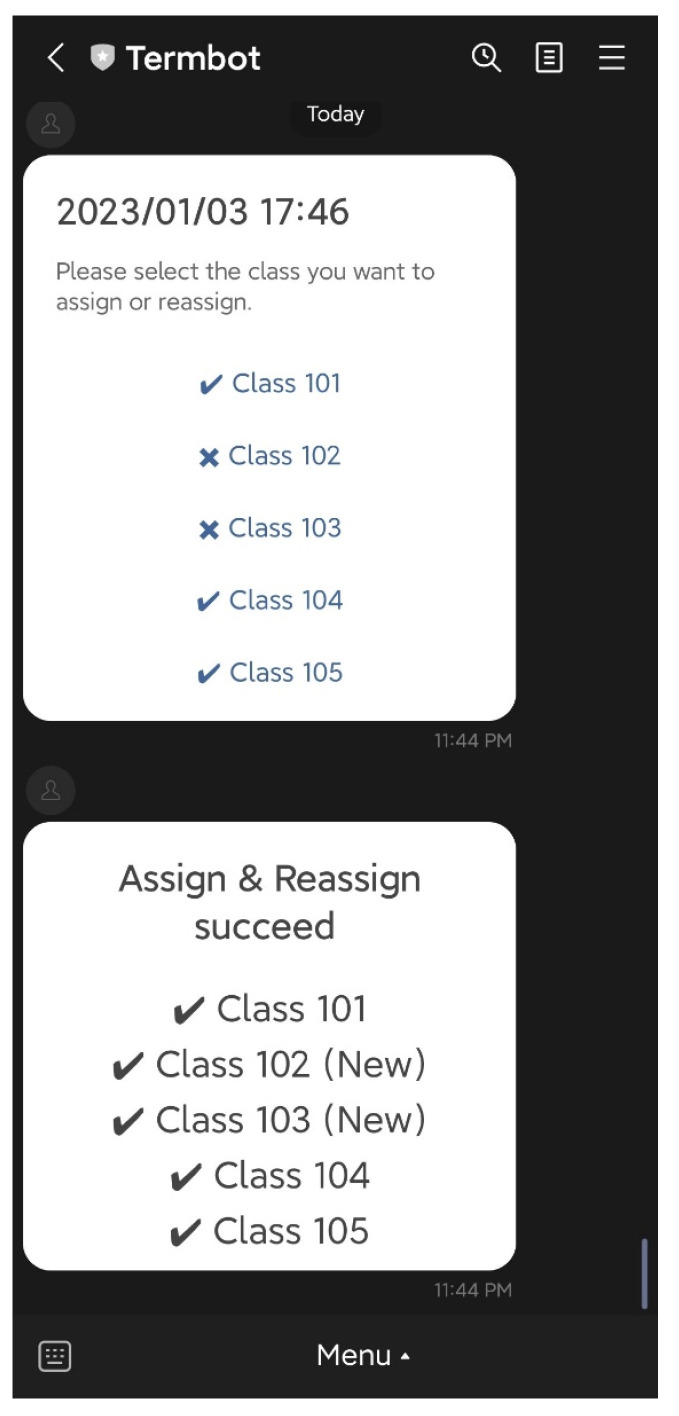
When the function “Assign & Reassign” ends.

**Figure 11 ijerph-20-04185-f011:**
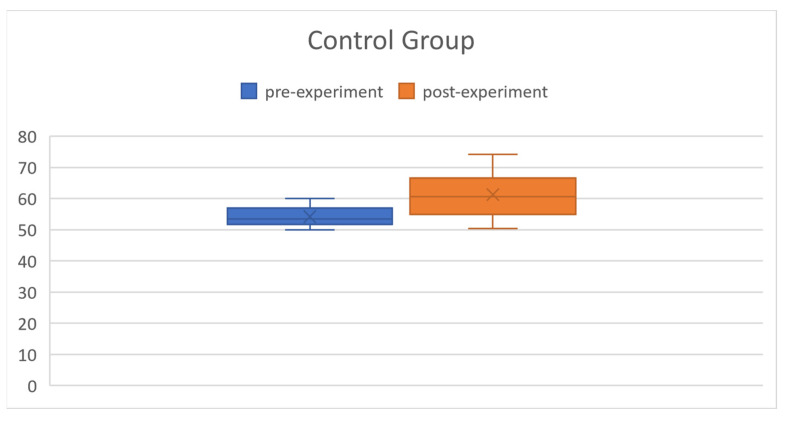
Pre-experiment and post-experiment data of the control group.

**Figure 12 ijerph-20-04185-f012:**
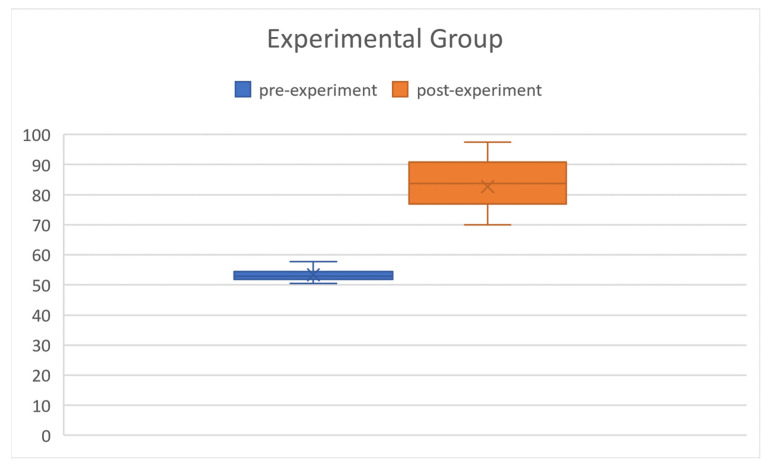
Pre-experiment and post-experiment data of the experimental group.

**Table 1 ijerph-20-04185-t001:** The SUS questionnaires.

	Questions
1	I think I’d be willing to use Termbot more often.
2	I think Termbot is too complicated.
3	I think Termbot is easy to use.
4	I think I need some help using Termbot.
5	I think the functionality of Termbot is well integrated.
6	I feel like Termbot has too many inconsistencies.
7	I can imagine that most people will learn to use Termbot very quickly.
8	I find Termbot cumbersome to use.
9	I am confident that I can use Termbot.
10	I need to learn a lot of extra information to use Termbot.

**Table 2 ijerph-20-04185-t002:** Before the experiment, *t*-tests for the experimental group and control group.

Group	Number	Average	Variation	*t*	*p*
Control Group	30	54.23	10.46	1.2749	0.2074
Experimental Group	30	53.33	4.54		

**Table 3 ijerph-20-04185-t003:** After the experiment, *t*-tests for the experimental group and control group.

Group	Number	Average	Variation	*t*	*p*
Control Group	30	61.31	54.41	−10.5072	0.0000
Experimental Group	30	82.69	69.90

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
