# Peer review of "Termbot: A Chatbot-Based Crossword Game for Gamified Medical Terminology Learning"

_ijerph, 2023, doi:10.3390/ijerph20054185_

Round 1

Reviewer 1 Report

The work is interesting. The authors explore Termbot as a chatbot-based online crossword game designed to help healthcare students learn medical terminology in a fun and convenient way. In this context students are using smartphones more and more, so Termbot was made to help them learn in a fun and easy way. With the crossword puzzles, Termbot turns boring medical terms into something fun and interesting. A controlled experiment was done to find out how Termbot affects how well students perform in school. The results showed, along the author explanations, that students trained with Termbot made significant progress in learning medical terms. In addition, the way Termbot turned learning into a game helped students learn more quickly and could be used in other fields.
Yet these contents of the work, which are explained from the authors there are some concerns on their work, which need to be addressed more.
I want to emphasize that the data used in the study must be verified to ensure the validity of the results. In this sense, the authors should check the data for outliers or other problems that may occur in the data. At the same time, authors should perform a sensitivity analysis to check the results and also consider other visual methods (e.g., comparing the performances using boxplots, etc.)

More clarity regarding data quality may be needed to ensure the study's final results. Also, there should be more information about the assumptions used in the study's methods. A more precise explanation of the methodology and data analysis process would significantly improve the results' robustness and the study's overall validity. To make the study more solid and reliable, it is essential to look into and deal with these issues in depth.

Author Response

Reply: Following your kind advice, boxplots have been added to Section 5.3 to illustrate the data distribution for this study. Furthermore, in Section 1, the relevant research question is also added to provide a clear focus for this study.

Reviewer 2 Report

In the paper, an interesting software, termbot, was designed. The organization of this paper is good and the novelty is obvious.

The review is as follows:

1.     Maybe a part named “related work” is needed to give the reader the background knowledge in detail.

2.     Try to compare your method with the state-of-the-art method(s) in the missing “performance evaluation” part.

3.     The termbot should be shown using a system flow chart, where each component of this system and the relationship between components should be described.

4.     I don’t know what the language is, which was used by the authors to design their termbot.

5.     As for the sus questionnaires, is it specific? Why do you set these specific questions? Can other questions apply?

6.     Figures 1,2,3, 4 should be placed in the middle.

7.     Tables in this paper should be organized in this journal’s form.

Author Response

  1. Maybe a part named “related work” is needed to give the reader the background knowledge in detail.

      Reply: A related work section has been added.

  1. Try to compare your method with the state-of-the-art method(s) in the missing “performance evaluation” part.

      Reply: Thanks for the valuable advice. Because "performance" may be affected by network speed factors, this study used the System Availability Scale (SUS) instead of "performance evaluation".

  1. The termbot should be shown using a system flow chart, where each component of this system and the relationship between components should be described.

      Reply: Thanks for the valuable advice. Figure 1 has been redrawn to describe the system and the relationship between components in more detail.

  1. I don’t know what the language is, which was used by the authors to design their termbot.

Reply: Python and JavaScript were used to design the Termbot. The relevant clarifications have been added to Section 3.3.

  1. As for the sus questionnaires, is it specific? Why do you set these specific questions? Can other questions apply?

      Reply: This study utilized the System Usability Scale (SUS), created by John Brooke in 1986 because the SUS is a widely used method for quickly assessing the usability of product system interfaces, desktop programs, and website interfaces. Therefore, it is very suitable for application in usability testing to evaluate the effectiveness of Termbot in meeting users' needs.

  1. Figures 1,2,3, 4 should be placed in the middle.

      Reply: Following your kind advice, those figures have been placed in the middle.

  1. Tables in this paper should be organized in this journal’s form.

      Reply: The Tables in this paper have been reorganized in this journal’s format.

Reviewer 3 Report

General:

-        Games are meant for entertainment and relaxing purposes. How do the authors think to encourage students taking up the app? Please include them in the discussion section.

Abstract:

-        The abstract is too long. Follow the mdpi guidelines.

Introduction:

-        Please shorten the introduction. The motivation is set in the first paragraph, then the later passages are too  much to compliment.

-        The authors mentioned “According to a 2020 report from Taiwan’s Ministry of Health and Welfare, there is over 337,942 medical staff who face similar difficulties in understanding and remembering a large number of medical terms.” Were these medical terms in local language or in other language like English? In general students struggle to learn words in different language.

-        “college students revealed they spend nearly 6 hours a day on their smartphones”. Same information is given in the abstract, please remove from one section.

-        A lot has been discussed on the role of crossword and chatbot in education. In the last paragraph only, TermBot operation is briefly discussed only. The individual role of chat bot in termbot is unclear. This will be confusing for readers not familiar with how chatbot operates.

2.3 Study instruments

-        The TermBot working principle figure 1 is unclear. Please show a complete schematic with the feedback mechanism.

2.4 System Schema:

-        Please use a table for what the notation signifies. Some information are repeated e.g. ID, Book etc.

3. TermBot:

-        The first paragraph is repeated information. Please remove it.

Discussion & future:

-        The discussion section is small. I think game puzzle like this is an excellent way to learn about misinformation. Use and cite the following articles to write few sentences on how tools like TermBot can be used for to combat health misinformation.

a)     Shams, A.B.; Hoque Apu, E.; Rahman, A.; Sarker Raihan, M.M.; Siddika, N.; Preo, R.B.; Hussein, M.R.; Mostari, S.; Kabir, R. Web Search Engine Misinformation Notifier Extension (SEMiNExt): A Machine Learning Based Approach during COVID-19 Pandemic. Healthcare 2021, 9, 156. https://doi.org/10.3390/healthcare9020156

b)     Health Chatbots for Fighting COVID-19: a Scoping Review. https://www.ncbi.nlm.nih.gov/pmc/articles/PMC7879453/

References:

-        Many paper cited are more than 10 years old. Please use more recent ones.

Author Response

General:

-Games are meant for entertainment and relaxing purposes. How do the authors think to encourage students taking up the app? Please include them in the discussion section.

Reply: Thanks for the valuable advice. Offering incentives and rewards were used to promote the usage of Termbot. The relevant clarifications have been added to the Discussion section. They are as follows:

Offering incentives and rewards: Providing incentives and rewards can motivate students to use Termbot and strive for success. These incentives and rewards could include certificates, prizes, or recognition.

Providing regular feedback: Regular feedback on students' progress can help to build their confidence and motivation to continue using Termbot. This feedback can also help to identify areas where they need to focus their attention and improve their learning outcomes.”

Abstract:

-The abstract is too long. Follow the mdpi guidelines.

Reply: The Abstract has been shortened to conform to MDPI specifications.

Introduction:

-1.1 Please shorten the introduction. The motivation is set in the first paragraph, then the later passages are too much to compliment.

Reply: Thanks for the valuable advice. The introduction has been shortened as you suggested

-1.2 The authors mentioned “According to a 2020 report from Taiwan’s Ministry of Health and Welfare, there is over 337,942 medical staff who face similar difficulties in understanding and remembering a large number of medical terms.” Were these medical terms in local language or in other language like English? In general students struggle to learn words in different language.

Reply: All medical terms mentioned in this study refer to English medical terms.

-1.3 “college students revealed they spend nearly 6 hours a day on their smartphones”. Same information is given in the abstract, please remove from one section.

Reply: This information in the abstract has been removed.

-1.4 A lot has been discussed on the role of crossword and chatbot in education. In the last paragraph only, TermBot operation is briefly discussed only. The individual role of chat bot in termbot is unclear. This will be confusing for readers not familiar with how chatbot operates.

Reply: Thanks for the valuable advice. In the Termbot Section, a detailed description of the operation steps for teachers and students has been added.

2.3 Study instruments

-The TermBot working principle figure 1 is unclear. Please show a complete schematic with the feedback mechanism.

Reply: Figure 1 has been redrawn to describe the system and the relationship between components in more detail. Additionally, the complete schematic with the feedback mechanism is shown in Figure 2.

2.4 System Schema:

-Please use a table for what the notation signifies. Some information are repeated e.g. ID, Book etc.

Reply: Thanks for the valuable advice. The relational diagram in Figure 2 has been redrawn as you suggested.

  1. TermBot:

-The first paragraph is repeated information. Please remove it.

Reply: The repeated information in the first paragraph has been removed.

Discussion & future:

-The discussion section is small. I think game puzzle like this is an excellent way to learn about misinformation. Use and cite the following articles to write few sentences on how tools like TermBot can be used for to combat health misinformation.

a)Shams, A.B.; Hoque Apu, E.; Rahman, A.; Sarker Raihan, M.M.; Siddika, N.; Preo, R.B.; Hussein, M.R.; Mostari, S.; Kabir, R. Web Search Engine Misinformation Notifier Extension (SEMiNExt): A Machine Learning Based Approach during COVID-19 Pandemic. Healthcare 2021, 9, 156. https://doi.org/10.3390/healthcare9020156

b)Health Chatbots for Fighting COVID-19: a Scoping Review. https://www.ncbi.nlm.nih.gov/pmc/articles/PMC7879453/

Reply: a) and b) related work has been included in the discussion section of this study.

References:

-Many paper cited are more than 10 years old. Please use more recent ones.

Reply: Some recent papers have been added to the references section.

Round 2

Reviewer 1 Report

The author have improved the work. Well done.

Reviewer 2 Report

this paper can be published in its current form.